# Probabilistic Riemannian Functional Map Synchronization for 3D Shape Correspondence

## Abstract

We consider the problem of graph-matching on a network of 3D shapes with uncertainty quantification. We assume that the pairwise shape correspondences are efficiently represented as *functional maps*, that match real-valued functions defined over pairs of shapes. By modeling functional maps between nearly isometric shapes as elements of the Lie group $SO(n)$, we employ *synchronization* to enforce cycle consistency of the collection of functional maps over the graph, hereby enhancing the accuracy of the individual maps. We further introduce a tempered Bayesian probabilistic inference framework on $SO(n)$. Our framework enables: (i) synchronization of functional maps as maximum-a-posteriori estimation on the Riemannian manifold of functional maps, (ii) sampling the solution space in our energy based model so as to quantify uncertainty in the synchronization problem. We dub the latter *Riemannian Langevin Functional Map (RLFM) Sampler*. Our experiments demonstrate that constraining the synchronization on the Riemannian manifold $SO(n)$ improves the estimation of the functional maps, while our RLFM sampler provides for the first time an uncertainty quantification of the results. Our implementation is publicly available.

## 1 Introduction

Many computer vision and medical imaging algorithms require the computation of shape correspondence (Van Kaick et al., 2011; Ma et al., 2021; Sahillioğlu, 2020), which is often a first step for applications ranging from texture transfer in computer graphics to segmentation of anatomical structures in computational medicine.

The problem of shape correspondence has traditionally been formulated as finding a matching between points defined on two three-dimensional shapes. Formally, consider two point sets $\mathbf{X} = \{\mathbf{x}_i\}_{i=1}^N \in \mathbb{R}^{M \times 3}$ and $\mathbf{Y} = \{\mathbf{y}_i\}_{i=1}^N \in \mathbb{R}^{N \times 3}$ associated with two shapes. Shape matching relates these point sets with a *partial permutation matrix* $\mathbf{P} \in \mathbb{P}^{M \times N}$ such that $\mathbf{X} = \mathbf{P}\mathbf{Y}$ and:

$$\mathbb{P}^{M \times N} = \{\mathbf{P} \in \{0,1\}^{M \times N} : \mathbf{P}\mathbf{1}_N = \mathbf{1}_M \ \wedge \ \mathbf{1}_M^\top \mathbf{P} \leq \mathbf{1}_N^\top\}, \tag{1}$$

where $\mathbf{1}_N$, $\mathbf{1}_M$ are an all-ones vector of length $N$ and $M$ respectively. Finding correspondences between the point sets $\mathbf{X}$ and $\mathbf{Y}$ amounts to optimize for $\mathbf{P}$, which is known to lie on the *monoid* (Arrigoni et al., 2017)—an object that complicates optimization procedures.

When $\mathbf{P}$ is a *total permutation matrix*, *e.g.* for $M = N$, the optimization problem can be relaxed. Shape matching can optimize for a *doubly stochastic matrix* (Douik & Hassibi, 2018; Birdal & Simsekli, 2019), while partial matching can be restricted to the set of *stochastic (multinomial) matrices* $\mathbb{S}^{M \times N}$:

$$\mathbb{S}^{M \times N} = \{\mathbf{S} \in \mathbb{R}^{M \times N} : S_{ij} > 0 \ \wedge \ \mathbf{S}\mathbf{1}_N = \mathbf{1}_M\}. \tag{2}$$

There are three fundamental challenges to estimating $\mathbf{S}$ of that form: (i) $\mathbf{S}$ is a relaxation of an entity $\mathbf{P}$ that is in nature discrete, (ii) $\mathbf{S}$ is generally not a bijection (neither doubly stochastic nor orthogonal) as the shapes can have different cardinality, (iii) increasing the point set cardinality quadratically increases the number of parameters to be estimated. Due to these challenges, performing optimization directly on the set of multinomial matrices $\mathbf{S}$ is also undesirable.

Hence, we instead consider the framework of *functional maps*. A functional map matches integrable real-valued functions $f$ and $g$ defined on the shapes $\mathcal{X}$ and $\mathcal{Y}$, instead of finding correspondences between the point sets $\mathbf{X}$ and $\mathbf{Y}$ (Ovsjanikov et al., 2012). To compute a functional map, we equip the two shapes $\mathcal{X}$ and $\mathcal{Y}$ with a reduced set of basis orthonormal real-valued functions $\mathbf{\Phi}_{\mathcal{X}}$ and $\mathbf{\Phi}_{\mathcal{Y}}$, corresponding to $\mathbf{X}$ and $\mathbf{Y}$ respectively. Then, by (Ovsjanikov et al., 2012) we can focus on finding a matrix $\mathbf{C}$ that is related to $\mathbf{P}$ through an area-weighted inner product with area $\mathbf{A}_{\mathcal{X}}$ as follows: $\mathbf{C} = \mathbf{\Phi}_{\mathcal{X}}^{\top} \mathbf{A}_{\mathcal{X}} \mathbf{P} \mathbf{\Phi}_{\mathcal{Y}}$. This is true for meshes and point clouds, since there the Laplacian discretization also involves mass and stiffness $\mathbf{C} = \mathbf{\Phi}_{\mathcal{X}}^{\top} \mathbf{P} \mathbf{\Phi}_{\mathcal{Y}}$. The matrix $\mathbf{C}$ is called the functional map.

In this framework, the correspondence for a pair of functions $f : \mathcal{X} \mapsto \mathbb{R}$ and $g : \mathcal{Y} \mapsto \mathbb{R}$ can be written as $\mathbf{Cb} = \mathbf{a}$, where $\mathbf{a}$ and $\mathbf{b}$ are the representation of $f$ and $g$ in the chosen bases $\mathbf{\Phi}_{\mathcal{X}}$ and $\mathbf{\Phi}_{\mathcal{Y}}$. Note that, if the bases are indicator functions, then $\mathbf{C} \equiv \mathbf{P}$. Yet, to reduce the representation complexity as well as to better handle the typical near-isometric mappings, it is common to use the first $N_B$ Laplace-Beltrami eigenfunctions (Ovsjanikov et al., 2012; Rodolà et al., 2019) as the bases. The correspondence for a set of pairs of functions $\{f_k\}_{k=1}^{K}$ and $\{g_k\}_{k=1}^{K}$ is then written as: $\mathbf{Cb}_k = \mathbf{a}_k$ for each pair indexed by $k$. The coefficients $a_k$'s and $b_k$'s can be conveniently stored in the columns of matrices $\mathbf{A}$ and $\mathbf{B}$. Solving the correspondence problem then amounts to minimizing an energy $E_1$ defined as:

$$E_1(\mathbf{C}) = \|\mathbf{CB} - \mathbf{A}\|^2, \tag{3}$$

which yields an optimal functional map $\mathbf{C}$ between the two shapes $\mathcal{X}$ and $\mathcal{Y}$. Eq. (3) is known as the *segment correspondence constraint* (Ovsjanikov et al., 2012). When the two shapes are nearly isometric, and the same number $N_B = n$ of Laplace-Beltrami eigenfunctions is chosen to describe the bases, then the functional map $\mathbf{C}$ can be modeled as an element of the square Stiefel manifold $St(n, n)$ (Chakraborty & Vemuri, 2019). For the sake of clarity, we focus on the component of $St(n, n)$ that is connected to the identity matrix. Consequently, we further model the space of functional maps as the Lie group $SO(n)$, *i.e.*, the special orthogonal (SO) group in $n$ dimensions representing rotations in $n$ dimensions. This means that the problem of minimizing the energy $E_1$ in Eq. (3) is an optimization problem constrained on $SO(n)$.

The functional map $\mathbf{C}_{ij}$ can be computed by solving the optimization problem from Eq. (3) on the two shapes $\mathcal{X}$ and $\mathcal{Y}$, now indexed by $i, j$. Considering a dataset of shapes, we can enhance the accuracy of the set of functional maps $\{\mathbf{C}_{ij}\}_{ij}$ through the process of synchronization Arrigoni et al. (2017); Birdal & Simsekli (2019); Huang et al. (2014); Cin et al. (2021). Synchronization uses the property of cycle consistency to improve on each functional map $\mathbf{C}_{ij}$. However, a naive optimizer cannot yield uncertainty quantification (UQ) for the problem at hand. In fact, to this date, there is no method to quantify uncertainties in the functional map synchronization. There either exist methods that enforce cycle consistency of functional maps without uncertainty quantification (optimizers), or methods that quantify the uncertainty of the synchronization process formulated with different representations such as $SO(3)$ Birdal et al. (2020) / $SE(3)$ Sun et al. (2019) or doubly stochastic matrices, as relaxations of permutations matrices Birdal & Simsekli (2019). However, we observed that these representations have practical shortcomings: point-based synchronization is not scalable to large point clouds and partial-permutations are hard to treat. Furthermore, most synchronization methods of functional maps do not take into account the geometric constraints that equip the space of the matrices $\{\mathbf{C}_{ij}\}_{ij}$, as being either $St(n, n)$ or $SO(n)$. Taking into account such geometric knowledge can assist the optimization problem and holds the promise to enhance the accuracy of the results. In this context, the contributions of this paper are as follows:

1. We develop a Riemannian Bayesian probabilistic framework to formulate the synchronization problem as a maximum-a-posteriori (MAP) estimation on the Lie group $SO(n)$ equipped with a bi-invariant Riemannian metric.

2. We introduce a Riemannian Langevin Functional Map (RLFM) sampler to perform uncertainty quantification associated with the synchronization of functional maps on $SO(n)$.

3. Our experiments show that the MAP estimation constrained on the Riemannian manifold enhances the accuracy of the maps and is robust in high noise regimes while our sampler can generate diverse solutions.

## 2 Related Work

Finding meaningful correspondences between two or more shapes is a well-studied area of computer vision, computer graphics and shape analysis. We review the topics closest to our method. Reviews of shape matching can be found in (Van Kaick et al., 2011; Sahillioğlu, 2020; Ma et al., 2021).

**Functional Maps** Functional maps were initially introduced in (Ovsjanikov et al., 2012) which studied the relationships between surfaces instead of point-to-point matching in Euclidean space. The original works have been extended in further research (Nogneng & Ovsjanikov, 2017; Rodolà et al., 2019; Corman et al., 2017; Yi et al., 2017; Melzi et al., 2019a; Donati et al., 2020).

A structured prediction model in the space of functional maps was designed in (Litany et al., 2017) to provide a compact representation of the correspondence between deformable 3D shapes. Learning techniques (Litany et al., 2017) of functional maps were extended in (Donati et al., 2020) to learn features from the 3D geometry rather than from some pre-computed descriptors, and by building a regularizer into the functional map computation layer to improve training speed. A dense correspondence between a set of 3D face surface meshes was computed with functional maps in (Zhang et al., 2016), together with a method that ensures cycle-consistency between maps. This work takes advantage of the lower dimension of functional maps in comparison to the traditional point set correspondences, and performs efficiently on very high resolution surface meshes. Functional map representation was employed in (Marin et al., 2020) to encode and infer shape maps throughout the registration process for non-rigid registration of 3D human shapes. Visual assessments of functional maps were discussed in (Melzi et al., 2019a) by introducing a visual evaluation method based on the transfer of the object-space normals across the two spaces. Characterizing distortion between pairs of shapes was tackled in (Corman et al., 2017) which extends a shape differences framework built on functional maps. Functional maps in the spectral domain were applied in (Yi et al., 2017) to synchronize unaligned eigenbases for the graph Laplacians to a common canonical space. The aligning functional maps succeed in encoding all the dual information on a common set of basis functions where global learning takes place. A probabilistic model for functional maps was introduced in (Rodolà et al., 2019) and used soft-maps with analytical distributions. In (Nogneng & Ovsjanikov, 2017), the isometries challenge of shape matching was tackled by employing functional maps corresponding to a point-wise map with diagonal descriptor matrices. A functional map synchronization algorithm was proposed in (Huang et al., 2021a) to jointly register multiple non-rigid shapes by synchronizing the maps relating learned functions defined on the point clouds. Gao et al. (2021) considered simultaneously synchronizing permutations and functional maps to increase the final accuracy. Similarly, Cao & Bernard (2022) proposed a data-driven approach for unsupervised learning of multi-graph matching.

**Map Synchronization** Map synchronization can be defined as the task of optimizing maps among a dataset of shapes to improve the maps computed between each pair of shapes. A tensor map synchronization was introduced in (Huang et al., 2019a) to establish high-quality correspondence maps across a heterogeneous shape collection. This work is based on a data-driven shape segmentation approach that utilizes maps to explore shape variabilities and identify meaningful shape decompositions into parts. The problem of jointly optimizing symmetry groups and pair-wise maps among a collection of symmetric objects was studied in (Sun et al., 2018). This work proposes a lifting map representation to encode both symmetry groups and maps between symmetry groups as well as a computational framework for joint symmetry and map synchronization. A supervised transformation synchronization approach was introduced in (Huang et al., 2019b). This work modifies a reweighted nonlinear least square approach and uses a neural network to automatically determine the input pairwise transformations and the associated weights. Similar research have then been conducted to solve scene flow and multi-body registration problems Gojcic et al. (2020); Huang et al. (2021b). A pioneer quantum algorithm was introduced in (Birdal et al., 2021) to solve a synchronization problem by focusing on permutation synchronization. This work involves solving a non-convex optimization problem in discrete variables. A ZoomOut technique was introduced in (Melzi et al., 2019b) to refine maps or correspondences by iterative upsampling in the spectral domain. Following the same idea, a consistent ZoomOut was proposed in (Huang et al., 2020), to efficiently refine correspondences among 3D shape collections, while promoting consistency in the resulting maps.

A probabilistic model for the permutation synchronization problem was introduced in (Birdal & Simsekli, 2019). This work minimizes a cycle consistency loss via a Riemannian-LBFGS algorithm and integrates a manifold-MCMC scheme that enables posterior sampling and thereby confidence estimation and uncertainty quantification. The paradigm of measure synchronization is introduced in (Birdal et al., 2020) for synchronizing graphs with measure-valued edges. This problem is formulated as maximization of the cycle-consistency in the space of probability measures over relative rotations. The aim is to estimate marginal distributions of absolute orientations by synchronizing the conditional ones, which are defined on the Riemannian manifold of quaternions. Synchronization of multi-graphs is addressed by (Cin et al., 2021), where multi-graphs are graphs with more than one edge connecting the same pair of nodes. The problem arises when multiple measures are available to model the relationship between two vertices. However, no probabilistic model of the synchronization problem has been introduced for the framework of functional maps.

**Contributions** Despite important advances on functional maps made in the aforementioned studies, to this date, there is no study on uncertainty quantification (UQ) for functional map synchronization. In this context, our main contribution is to develop a probabilistic inference paradigm to perform synchronization on functional maps while taking into account the geometry of the space they belong to. To the best of our knowledge, this paper is the first to deploy a Riemannian MCMC sampler to achieve this goal.

## 3 Synchronization of Functional Maps

In this section, we explain how to refine functional maps in a synchronization scenario that enforces cycle consistency between maps.

**Synchronization as a Constrained Optimization** Consider a dataset of shapes $\mathcal{X}_i$'s, for $i = 1, \dots, N_s$ that are organized into a directed graph with nodes $\{1, \dots, N_s\}$ and edges $\mathcal{E} \subset \{1, \dots, N_s\} \times \{1, \dots, N_s\}$. Consider pairwise functional maps $\mathbf{C}_{ij}$'s that have been computed for each edge $(i, j) \in \mathcal{E}$. We further assume the existence of a canonical universe defining a canonical shape to which every other shape maps to. Without loss of generality, we consider $\mathcal{X}_1$ as the canonical shape[1]. *Functional map synchronization* is then interested in finding the underlying *absolute functional maps* $\mathbf{C}_i$ for $i \in \{1, \dots, N_s\}$ with respect to the common origin (*e.g.*, $\mathbf{C}_1 = \mathbf{I}$, the identity matrix) that respects the cycle consistency of the underlying graph structure:

$$\mathbf{C}_i \mathbf{C}_{ij} = \mathbf{C}_j. \tag{4}$$

This equation echoes the factorization of the (relative) functional maps as $\mathbf{C}_{ij} \approx \mathbf{C}_i^+ \mathbf{C}_j$ (see (Huang et al., 2014)), where $+$ denotes the pseudo-inverse. The cycle consistency constraint gives rise to the optimization problem of functional maps synchronization:

$$\underset{\{\mathbf{C}_i\}_{i=1}^{N_s}}{\arg\min} \sum_{(i,j) \in \mathcal{E}} \|\mathbf{C}_i \mathbf{C}_{ij} - \mathbf{C}_j\|_F^2, \tag{5}$$

where $\| \cdot \|_F$ is the Frobenius norm and we optimize for $N_s$ absolute functional maps. An unconstrained optimization problem can solve for $\mathbf{C}_i \in \mathbb{R}^{n \times n}$, where $n$ is the number of Laplace-Beltrami eigenfunctions presented in intruction. Yet, it is preferable to take into account the geometric constraints defining the properties of functional maps. Consequently, it is possible to minimize Eq. (4) via Riemannian optimization algorithms as done in (Ovsjanikov et al., 2012) by constraining each functional map $\mathbf{C}_i$ to live on the manifold of area-preserving functional maps $\mathcal{F} \subset \mathbb{R}^{n \times n}$. Due to the orthonormality of the columns of $\mathbf{C}$, it is now common to treat $\mathcal{F}$ to be the *Stiefel manifold*. By further restricting our study to the functional maps belonging to the component of the manifold connected to the identity matrix, we model $\mathcal{F}$ as the special orthogonal group:

$$\mathcal{F} = SO(n) = \{\mathbf{C} \in \mathbb{R}^{n \times n} : \mathbf{C}^\top \mathbf{C} = \mathbf{I}_n, \det(\mathbf{C}) = 1\}, \tag{6}$$

---

[1]Note that the entire synchronization problem is subject to *gauge freedom*, a solution is determined only up to a global isometry (Birdal et al., 2021).

where $\mathbf{I}_n$ is the $n \times n$ identity matrix. The space of functional maps $\mathcal{F}$ is thus represented by the Lie group $SO(n)$ of rotations in $n$-dimensions. The constrained synchronization problem is then written as:

$$\underset{\{\mathbf{C}_i \in SO(n)\}_{i=1}^{N_s}}{\arg\min} \sum_{(i,j) \in \mathcal{E}} \|\mathbf{C}_i \mathbf{C}_{ij} - \mathbf{C}_j\|_F^2. \tag{7}$$

**Computations and Optimization on** $SO(n)$  The Lie group $SO(n)$ can be naturally equipped with a structure of Riemannian manifold. That is, we do not rely Euclidean linear operations to compute with elements of $SO(n)$. Instead, we use operations that respect the Lie group structure through the definition of a Riemannian metric that is bi-invariant on $SO(n)$, *i.e.*, that turns $SO(n)$ into a Riemannian manifold while respecting intrinsic algebraic properties of the Lie group (Miolane & Pennec, 2015; Boumal, 2020). As a result, the usual operation from Euclidean space that consists in "adding a vector to a point" can be generalized by the retraction operator $R_\mathbf{C}$ at point $\mathbf{C} \in SO(n)$ that adds a tangent vector to $SO(n)$ to a given point $\mathbf{C} \in SO(n)$. To project a vector $\mathbf{V}$ to a tangent vector at a given point $\mathbf{C} \in SO(n)$, we also define the projection operator $\Pi_\mathbf{C}$. Specifically, these operators can be defined as (Roy et al., 2019; Edelman et al., 1998):

$$\Pi_\mathbf{C}(\mathbf{V}) = \mathbf{V} - \mathbf{C}\,\mathrm{sym}(\mathbf{C}^\top \mathbf{V}), \tag{8}$$

$$R_\mathbf{C}(\boldsymbol{\xi}_\mathbf{C}) = \mathrm{qf}(\mathbf{C} + \boldsymbol{\xi}_\mathbf{C}), \tag{9}$$

where $\mathbf{V} \in \mathbb{R}^{n \times n}$ is a matrix in the ambient Euclidean space, $\mathrm{sym}(\mathbf{A}) = \frac{1}{2}(\mathbf{A} + \mathbf{A}^\top)$ and $\mathrm{qf}(\cdot)$ is the adjusted $\mathbf{Q}$ factor of the $\mathbf{QR}$-decomposition. In practice, to obtain $\mathrm{qf}(\cdot)$, one performs $\mathbf{QR}$-decomposition followed by sign-swaps on all the columns whose corresponding diagonal elements in $\mathbf{R}$ are negative.

## 4   Probabilistic Synchronization of Functional Maps

Instead of minimizing Eq. (5) or its Riemannian version Eq. (7) as done in the literature, we propose to characterize the empirical posterior distribution related to the energy—in an approach similar to (Birdal & Simsekli, 2019). We formulate the functional map synchronization problem in a probabilistic fashion. We treat the pairwise relative functional maps $\mathbf{C}_{ij}$'s as observed random variables and the absolute functional maps $\mathbf{C}_i$'s as latent random variables. The probabilistic model will enable us to cast the synchronization problem as an inference problem.

**Probabilistic Model**  We consider the case where we can define an analytical probability distribution on functional maps, whose mode can be set to a given map. This case covers the practically important scenario where probability distributions are modelled as Gaussians on the Riemannian manifold $\mathcal{F} = SO(n)$ (Pennec, 2006). Specifically, we assume that the observed relative functional maps $\mathbf{C}_{ij}$'s are generated by a probabilistic model that has the following hierarchical structure:

$$\mathbf{C}_i \sim p(\mathbf{C}_i) \qquad \mathbf{C}_{ij} \sim p(\mathbf{C}_{ij} \,|\, \mathbf{C}_i, \mathbf{C}_j) \qquad \forall i,j \in \{1, \dots, N_s\}, \tag{10}$$

where the latent variables $\mathbf{C} \equiv \{\mathbf{C}_i\}_{i=1}^{N_s}$ denote the true values of the absolute maps with respect to a common origin, $\mathcal{C} \equiv \{\mathbf{C}_{ij}\}_{i,j=1}^{N_s}$ are the observed relative maps, $p(\mathbf{C}_i)$ refers to the *prior distribution* of each latent variable, and in a vein similar to Birdal et al. (2018), $p(\mathbf{C}_{ij} \,|\, \mathbf{C}_i, \mathbf{C}_j)$ is the *generative model* of the observations.

Expecting that the true relative map is close to the expected value (corresponding to the mode for Gaussians) up to a flexibility determined by $\boldsymbol{\Lambda}$, we have the following property:

$$\underset{\mathbf{C}_{ij}}{\arg\max} \left\{ p(\mathbf{C}_{ij} | \mathbf{C}_i, \mathbf{C}_j) \right\} = \underset{\mathbf{C}_{ij}}{\arg\max} \left\{ P(\mathbf{C}_i^+ \mathbf{C}_j, \boldsymbol{\Lambda}) \right\} = \mathbf{C}_i^+ \mathbf{C}_j. \tag{11}$$

where $\mathbf{C}^+$ denotes the pseudo-inverse and $P$ refers to the distribution whose mode and concentration are $\mathbf{C}_i^+ \mathbf{C}_j$ and $\boldsymbol{\Lambda}$, respectively. This generative modelling strategy sets the most likely value of the relative pose to the deterministic value $\mathbf{C}_i^+ \mathbf{C}_j$.

**Inference**  The problem of probabilistic synchronization of functional maps becomes two-fold.

- Problem 1: Find the maximum a-posteriori (MAP) estimate. The MAP estimate $\mathbf{C}^\star$ is given by:

$$\mathbf{C}^\star = \underset{\mathbf{C} \in \mathcal{F}^{N_s}}{\arg\max}(\log p(\mathbf{C} \,|\, \mathcal{C})) = \underset{\mathbf{C} \in \mathcal{F}^{N_s}}{\arg\max} \left( \sum_{(i,j) \in \mathcal{E}} \log p(\mathbf{C}_{ij} | \mathbf{C}_i, \mathbf{C}_j) + \sum_{i=1}^{N_s} \log p(\mathbf{C}_i) \right), \tag{12}$$

  which follows from Bayes Theorem (see Appendix A).

- Problem 2: Find the full posterior distribution $p(\mathbf{C} \,|\, \mathcal{C})$. The posterior distribution of the absolute maps is written as: $p(\mathbf{C} \,|\, \mathcal{C}) \propto p(\mathcal{C} \,|\, \mathbf{C}) \times p(\mathbf{C})$. Beyond computing the MAP, we are interested in the full posterior distribution on $\mathbf{C}$ to provide uncertainty quantification for the absolute maps.

We note that specific assumptions in our probabilistic model can recover instances of the synchronization problem as performed in the literature. First, assuming a uniform prior distribution over the absolute maps reformulates MAP into a *Maximum Likelihood Estimation (MLE)* problem, as shown below:

$$\mathbf{C}^\star = \underset{\mathbf{C} \in \mathcal{F}^{N_s}}{\arg\max} \sum_{(i,j) \in \mathcal{E}} \log p(\mathbf{C}_{ij} | \mathbf{C}_i, \mathbf{C}_j). \tag{13}$$

Next, assuming an entropy-maximizing Gaussian distribution on the Riemannian manifold $\mathcal{F} = SO(n)$ for the generative model, with a mean $\mu = \mathbf{C}_i^+ \mathbf{C}_j$ and concentration matrix equal to the identity (Pennec, 2006), gives:

$$\mathbf{C}^\star = \underset{\mathbf{C} \in \mathcal{F}}{\arg\max} \sum_{(i,j) \in \mathcal{E}} \log \left( k \exp \left( -\frac{(\log_\mu \mathbf{C}_{ij})^T \log_\mu \mathbf{C}_{ij}}{2} \right) \right) = \underset{\mathbf{C} \in \mathcal{F}}{\arg\max} \sum_{(i,j) \in \mathcal{E}} \log \left( k \exp \left( -\frac{\text{dist}^2(\mathbf{C}_{ij}, \mu)}{2} \right) \right) \tag{14}$$

where log denotes the Riemannian logarithm (Pennec, 2006) and dist the Riemannian geodesic distance for the bi-invariant Riemannian metric on $\mathcal{F} = SO(n)$. We also note that we can drop the normalization constant $k$ of the probability density function that does not depend on $\mathbf{C}$.

Interestingly, Eq. (14) recovers the optimization problem from Eq. (5) if we consider an Euclidean geometry instead of a Riemannian geometry for $\mathcal{F} = SO(n)$. In the Euclidean case, the Riemannian logarithm becomes: $\log_\mu \mathbf{C}_{ij} = \mathbf{C}_{ij} - \mu = \mathbf{C}_{ij} - \mathbf{C}_i^+ \mathbf{C}_j$. This yields:

$$\mathbf{C}^\star = \underset{\mathbf{C} \in \mathcal{F}^{N_s}}{\arg\max} \sum_{(i,j) \in \mathcal{E}} -||\mathbf{C}_{ij} - \mathbf{C}_i^+ \mathbf{C}_j||^2 = \underset{\mathbf{C} \in \mathcal{F}^{N_s}}{\arg\min} \sum_{(i,j) \in \mathcal{E}} ||\mathbf{C}_{ij} - \mathbf{C}_i^+ \mathbf{C}_j||^2. \tag{15}$$

**Riemannian Langevin Functional Map (RLFM) Sampler** The inference problems 1-2 above are challenging in practice and cannot be directly addressed by standard methods such as gradient descent (problem 1) or standard MCMC methods (problem 2). The difficulty primarily originates from the facts that: (i) the posterior density is non-log-concave (*i.e.*, the negative log-posterior is non-convex); and (ii) any algorithm that aims to solve one of these problems should be able to operate in the particular manifold of this problem. To tackle the inference problems, we adopt an algorithmically simple yet effective algorithm: the *Riemannian Langevin Monte Carlo* explained in (Birdal & Simsekli, 2019; Girolami & Calderhead, 2011) and adapted to our framework into a *Riemannian Langevin Functional Map (RLFM) Sampler* as detailed below. Compared to methods deployed in the literature, our MC sampler uniquely provides a measure of uncertainty associated with the correspondences between two shapes after synchronization, as it samples from the posterior distribution $p(\mathbf{C} \,|\, \mathcal{C})$ of the functional maps.

We express the posterior of interest $p(\mathbf{C} \,|\, \mathcal{C})$ as a density $\pi_\mathcal{H}(\mathbf{C})$ with respect to the *Hausdorff measure* $\mathcal{H}$ written as:

$$\pi_\mathcal{H}(\mathbf{C}) \triangleq p(\mathbf{C} \,|\, \mathcal{C}) \propto \exp(-U(\mathbf{C})), \tag{16}$$

where $U$ is called the *potential energy* and has the following form:

$$U(\mathbf{C}) \triangleq -(\log p(\mathcal{C} \,|\, \mathbf{C}) + \log p(\mathbf{C})). \tag{17}$$

Assuming a uniform prior $p(\mathbf{C})$ and a Riemannian Gaussian distribution for the generative model $p(\mathcal{C} \,|\, \mathbf{C})$, we derive the potential energy $U(\mathbf{C})$ for the Langevin sampler as follows:

$$U(\mathbf{C}) = \sum_{(i,j) \in \mathcal{E}} \text{dist}^2(C_i^+ C_j, C_{ij}) + \text{constant}, \tag{18}$$

where dist is the Riemannian distance associated with the bi-invariant Riemannian metric on $\mathcal{F} = SO(n)$.

Next, we define a *smooth embedding* $\xi : (\mathbb{R}^{n \times n})^{N_s} \mapsto \mathcal{F}$ such that $\xi(\tilde{\mathbf{C}}) = \mathbf{C}$. If we consider the embedded posterior density $\pi_\lambda(\tilde{\mathbf{C}}) \triangleq p(\tilde{\mathbf{C}} | \mathcal{C})$ with respect to the Lebesgue measure $\lambda$, then the area formula from Theorem 1 in (Diaconis et al., 2013) gives:

$$\pi_{\mathcal{H}}(\mathbf{C}) = \pi_\lambda(\tilde{\mathbf{C}}) / \sqrt{|\mathbf{G}(\tilde{\mathbf{C}})|}, \tag{19}$$

where $|\mathbf{G}(\tilde{\mathbf{C}})|$ denotes the determinant of the Riemann metric tensor $\mathbf{G}$ at $\tilde{\mathbf{C}}$. We can then consider the following stochastic first order differential equation (SDE) similar to the one in (Birdal & Simsekli, 2019):

$$d\tilde{\mathbf{C}}_t = (-\mathbf{G}^{-1}\nabla_{\tilde{\mathbf{C}}} U_\lambda(\tilde{\mathbf{C}}_t) + \mathbf{\Gamma}_t)dt + \sqrt{2/\beta \mathbf{G}^{-1}}d\mathbf{B}_t, \tag{20}$$

where $t$ denotes time, $\mathbf{B}_t$ refers to the Brownian motion, and $\beta$ is an hyperparameter. Additionally, $\mathbf{\Gamma}_t$ is called the correction term and is defined as follows: $\left[\mathbf{\Gamma}_t(\tilde{\mathbf{C}})\right]_i = \sum_{j=1}^{N_s n^2} \partial \left[\mathbf{G}_t^{-1}(\tilde{\mathbf{C}})\right]_{ij} / \partial \tilde{\mathbf{C}}_j$.

Following a similar line of thought to (Birdal & Simsekli, 2019), we can now develop our integrator. However, in contrast to the *retraction Euler integrator* developed in (Birdal & Simsekli, 2019), we propose a direct *geodesic integrator* that leverages the analytical expression of the Riemannian exponential map that is used as the retraction $R_{\mathbf{C}}$. Specifically, we propose to simulate the SDE as follows:

$$\mathbf{V}_i^{(k+1)} = \Pi_{\mathbf{C}_i^{(k)}}(h\nabla_{\mathbf{C}_i} U(\mathbf{C}_i^{(k)}) + \sqrt{2h/\beta}\mathbf{Z}_i^{(k+1)}) \tag{21}$$

$$\mathbf{C}_i^{(k+1)} = R_{\mathbf{C}_i^{(k)}}(\mathbf{V}_i^{(k+1)}), \qquad \forall i \in \{1, \dots, N_s\}, \tag{22}$$

where $h > 0$ denotes the step-size, $k$ denotes the iterations, $\mathbf{Z}_i^{(k)}$ denotes standard Gaussian random variables in $\mathbb{R}^{n \times n}$, $\{\mathbf{C}_i^{(0)}\}$ denotes the initial absolute functional maps at $k = 0$. After warm-up of the Markov Chain, this process generates samples $\mathbf{C}$ according to the posterior distribution $p(\mathbf{C} | \mathcal{C})$. As proven in Birdal & Simsekli (2019), this integrator is identical to the Riemannian Langevin algorithm on $SO(N)$, which is proven to converge at a geometric rate for densities defined on the manifold and satisfying a Log-Sobolev inequality (Wang et al., 2020).

## 5 Experiments

We detail the experimental set up, including datasets and implementation details, that will demonstrate the applicability and advantages of the proposed method.

**Shape Dataset** For our baseline experiments, we use the 11 cat shapes from the TOSCA dataset (Bronstein et al., 2006; 2007; TOS, 2010). This dataset showcases a wide variety of poses and shapes which makes it suitable for baseline testing. Moreover, shape meshes in this dataset have the same triangulation and vertices indexed accordingly —which can be used as a per-vertex ground truth correspondence in correspondence experiments.

**Graph Generation** We organize the cat shapes into various graphs with at most 11 nodes, generated via a random graph generator using NetworkX (NetworkX, 2021). In these graphs, the nodes correspond to $N_s$ cat shapes selected among the 11 available in the TOSCA dataset and the edges depict the correspondence relationships between the shapes. We consider node count $N_s$ (number of shapes) and edge density as hyperparameters in our system. This allows us to evaluate the performance of our method based on the number of shapes, the connectivity, and the number of known initial correspondences (or initial functional maps) between the shapes.

**Ground Truth Functional Maps**  We consider one graph of cat shapes. For each pair $(i, j)$ of cats within the graph, *i.e.* for each edge, we compute the ground truth relative functional maps $\mathbf{C}_{ij}$ using the original implementation in (Ovsjanikov et al., 2012) and adapting the source code from (Maks et al., 2017). We use $n = 20$ basis functions on the source cat shape and $n = 20$ basis functions on the target cat shape, so that the functional maps are of size $20 \times 20$. Following Eq. (4), we generate the ground truth absolute functional maps $\mathbf{C}_i$ from the relative maps $\mathbf{C}_{ij}$ by performing a depth first search on the graph of cat shapes, and restricting the solutions to the Lie group $SO(n)$.

**Perturbations**  In practice, we do not observe ground-truth relative nor absolute functional maps. Instead, we need to estimate the absolute functional maps from noisy relative functional maps. To simulate noisy relative functional maps, we induce synthetic perturbations on the ground truth functional maps with different noise levels. Specifically, we add a Gaussian perturbation to the initial functional maps and vary the variance $\sigma^2 \in \{0.1, 0.2, \ldots, 1.0\}$ and project the resulting corrupted matrices on $SO(n)$ to get *corrupted relative functional maps*, denoted as $\mathbf{C}_{ij}^{\mathrm{corr}}$, which represent the observations of our probabilistic model. These perturbations will allow us to showcase the efficiency of our algorithm under varying levels of corruption $\sigma^2$.

**Riemannian Computation and Optimization**  Both the optimization (problem 1) and MCMC sampling (problem 2) are performed on the product manifold (Şuhubi, 2013) $SO(n)^{N_s}$ where each $SO(n)$ component represents the underlying manifold of one absolute map $\mathbf{C}_i$, for $i = 1, \ldots, N_s$. For the optimization, assuming a uniform or uninformative prior over the functional maps, we use Riemannian Trust-Regions (Boumal, 2015; Absil et al., 2007) to find the solution of the MLE (Maximum Likelihood Estimation) given by Eq. (5). The implementation of our methods uses the Riemannian optimization library Pymanopt (Townsend et al., 2016) and Riemannian geometry library Geomstats (Miolane et al., 2020).

## 6 Results

In this section, we provide a quantitative analysis of our methods and compare their performances using the TOSCA dataset (see Section 5). For consistency, the results in this section are reported on graphs with $N_s = 4$ cat shapes. We measure the connectivity of these graphs based on the number of edges present —which is referred to as *edge density*. For example, an edge density of 100% means that the graph is fully connected.

### 6.1 Baselines: Synchronization without Uncertainty Quantification

In this subsection, we evaluate the baseline methods that performs functional maps synchronization without uncertainty quantification, as is commonly done in the literature, using MLEs from Eq. (7). Depending on the optimization constraints, we consider two variations:

1. **Unconstrained: Euclidean MLE** (1)**.** The optimization of the objective from Eq. (7) is not constrained and is performed in the space $\mathbb{R}^{n \times n}$ for each $\mathbf{C}_i$.
2. **Constrained: Riemannian MLE** (2)**.** The optimization of the objective from Eq. (7) is constrained over the Lie group $SO(n)$ for each $\mathbf{C}_i$.

To benchmark performance in each of these variations, we run the optimization independently 10 times and calculate the Fréchet mean of the estimates $\mathbf{C}_i^\star$. This ensures that the final result is independent of the initialization. The constrained MLE is expected to perform better than the unconstrained MLE because it abides by the underlying geometric constraints and properties of the functional maps.

We evaluate the two variations of the baseline for observed relative functional maps with varying levels of noise. Figure 1 (left) shows the Frobenius distance between the MLEs and the corresponding ground truth during optimization for the unconstrained MLE (1) (full line) and the constrained MLE (2). The results obtained from constrained optimizations outperform the unconstrained ones. We observe that both methods converge quickly, typically within 25 iterations, and both refine the corrupted observed functional maps to functional maps closer to their ground truth values. The distances to the ground truth after convergence are reported in Table 1 of the next Section. Figure 1 (right) illustrates one estimated functional maps from MLE (2) visualized as vertex-to-vertex correspondence between a source shape (cat 4) and a target shape (cat 10). We observe that the anatomies of the cats are matched properly.

Convergence of MLEs

Estimated Functional Map

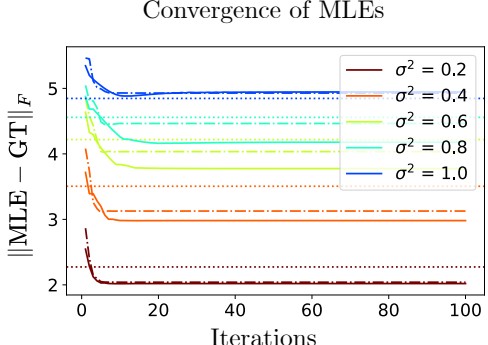

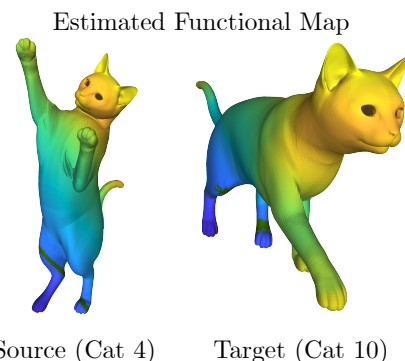

Source (Cat 4)     Target (Cat 10)

Figure 1: **Baselines, *i.e.,* synchronization without uncertainty quantification using Maximum Likelihood estimators (MLEs). Left:** Convergence of the MLE estimators with constraints on the Lie group $SO(n)$ (full lines) or without (dashed lines). The color of the each curve represents the variance of the noise $\sigma^2$ for the observed (corrupted) relative functional maps $\mathbf{C}_{ij}^{\text{corr}}$. The horizontal dotted lines represents the distance of the $\mathbf{C}_{ij}^{\text{corr}}$ to the ground truth, averaged over the $i, j$ in $\mathbf{C}_{ij}$. This Figure shows the plot for a graph with $N_s = 4$ nodes (shapes) and an edge density of $D = 100\%$ corresponding to a fully connected graph. **Right:** Vertex-to-vertex correspondence from TOSCA Cat shape 4 to Cat shape 10 (right image) using the estimated functional map from the constrained MLE. The source vertices are mapped to target vertices with the same color.

## 6.2 MCMCs: Synchronization with Uncertainty Quantification

In this subsection, we evaluate the results generated by our Riemannian Langevin Functional Map (RLFM) Sampler developed in Eq. (21) and Eq. (22). Compared to methods deployed in the literature, our sampler uniquely provides a measure of uncertainty associated with the correspondences between two shapes after synchronization—as it generates functional maps samples from the posterior $p(\mathbf{C} \,|\, \mathcal{C})$. We first illustrate uncertainty quantification, together with the quality of the sampled functional maps. Next we quantitatively evaluate the quality of the MAP estimator and the sampler.

**Uncertainty Quantification and Qualitative Evaluation of the Correspondences** Figure 2 shows functional maps samples generated by our MCMC. Specifically, we show how a specific vertex on one (source) cat shape is mapped on another (target) cat shape using 100 functional maps drawn for the RLFM. We use two examples with two source cat shapes (cats 9 and 1) and two target cat shapes (cats 5 and 2 respectively). In each source shape, we highlight the specific vertex of interest: vertices #9000 and #10000 for the left and right sides of Figure 2 respectively. For each functional map sampled from the MCMC, we calculate the vertex corresponding to the fixed vertex in the target cat shapes. Using 100 samples provides a distribution of correspondences: high density corresponds to lighter (yellow) colors while low density corresponds to darker (blue) colors in the colormap in Figure 2. By contrast with traditional synchronization methods, our approach provides a density distribution of correspondences, that is interpreted as a measure of uncertainty for the correspondence for the source vertices. Figure 2 also illustrates the quality of the samples drawn from MCMC, using 100 samples. We find that the Langevin functional maps samples map the source vertices to positions that are close to the actual corresponding vertex position in the target shapes. Moreover, many of the Langevin samples point to the same vertex as well, which illustrates the stability of the samples.

**Oracle Method for Evaluating Sampler Accuracy** We evaluate the results generated by our Langevin sampler by collecting its "best" functional maps through an oracle approach. We consider the Riemannian Langevin Functional Map Sampler with Euclidean distance in the potential energy $U$, *i.e.*, MC (1). Specifically, for each entry $c$ in the ground truth functional map $\mathbf{C}_i$, we iterate over all the samples $\tilde{\mathbf{C}}_i$ generated from the sampler and find the entry $\tilde{c}$ closest to $c$. Repeating this over all the entries of $\mathbf{C}_i$ gives a new functional map $\hat{\mathbf{C}}_i$ where the entries are the different $\tilde{c}$. The functional map $\hat{\mathbf{C}}_i$ represents the "best" possible map that our sampler can generate. We measure the Frobenius norm similarity score, which is an accuracy

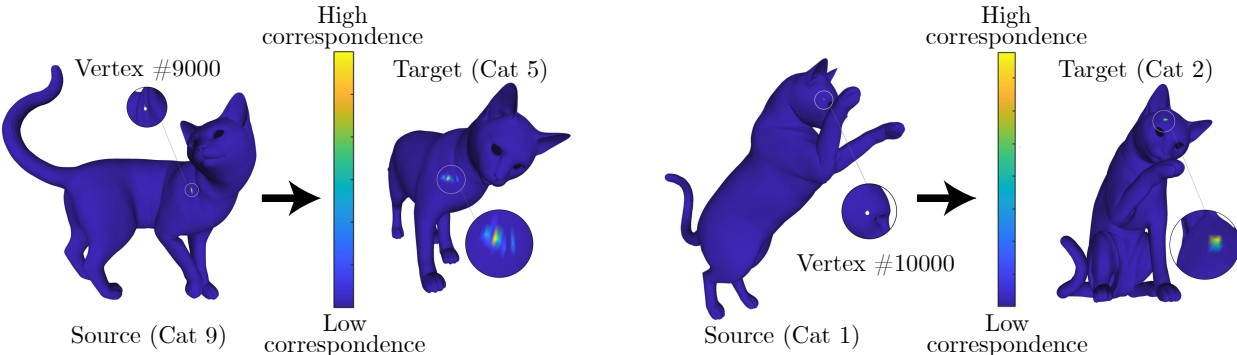

Figure 2: **Vertex correspondence using 100 samples of Riemannian Langevin Functional Map (RLFM) Sampler for two pairs of source and target shapes.** The source vertex positions have been marked with white dots for visibility. The target images show the vertex mappings using the computed functional maps. The value of the colormap define how many samples have established correspondence with the source vertex on each target vertex.

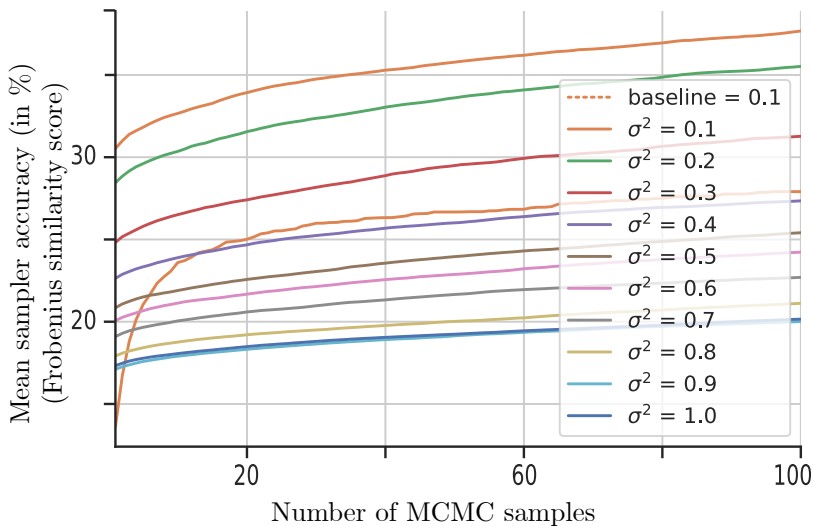

Figure 3: **Accuracy for Riemannian Langevin Functional Map (RLFM) Sampler with Euclidean Distance as Potential Energy MC (1).** The X-axis represents the number of samples generated from the Langevin sampler. The colors of the curve represents the levels of noise applied to generate the corrupted observed functional maps. The accuracy is calculated using the oracle method described in Subsection 6.2, which includes a description of the baseline (dotted line). The results are shown for an edge density $D$ of 100%.

score represented as:

$$\text{Acc}(\mathbf{C}_i, \hat{\mathbf{C}}_i) = \frac{1}{1 + \|\mathbf{C}_i - \hat{\mathbf{C}}_i\|_{\text{F}}} \qquad \forall i \in 1, \ldots, N_s, \tag{23}$$

between the ground truth functional map $\mathbf{C}_i$ with the newly constructed "best" map $\hat{\mathbf{C}}_i$. We calculate this accuracy for different numbers of Langevin samples and report the results in Figure 3 for an edge density $D$ of 100%. As expected, the similarity score increases with the increasing number of samples and decreases with an increasing value of corruption noise. As a baseline, we repeat the procedure using uniformly distributed random samples from $SO(n)$ with the first entry as the point estimate from maximum likelihood in place of samples from the posterior. This is shown as a dotted line in Figure 3, which is largely outperformed by the proposed sampler.

**Quantitative evaluation** We evaluate the results of our proposed RLFM estimation method quantitatively and compare them with the MLE baselines. As we did for the quantitative evaluations of the MLE baselines, we consider two variations for the MCs:

1. **Euclidean MC** (1)**.** The Euclidean distance is used to defined the potential energy $U$ of the Langevin Monte Carlo sampler in Eq. (18).
2. **Riemannian MC** (2)**.** The Riemannian geodesic distance is used to define the potential energy $U$ of the Langevin Monte Carlo sampler in Eq. (18).

We generate 1000 samples through the RLFM sampler and then calculate the Fréchet mean of the functional maps. The mean functional map is then compared to the ground truth functional map using the Frobenius norm as reported in Table 1 and Figure 4, which allows use to compare the accuracy of our estimators to the MLE baselines from the previous subsection. In the Euclidean variation that generates samples using Euclidean Langevin MCMC, we project the functional maps back onto the Lie group $SO(n)$. We compare the results for the MLE baselines as well as for our proposed MC methods for varying noise levels on the observed functional maps. We consider graphs with $N_s = 11$ nodes (cat shapes) and varying graph edge density $D$, defined as the sum of the degrees of all nodes in the graph compared to the total possible degrees in the corresponding fully connected graph.

| | | | Noise Levels ($\sigma^2$) | | | | |
|---|---|---|---|---|---|---|---|
| | Functional Maps | Edge Density | 0.2 | 0.4 | 0.6 | 0.8 | 1.0 |
| Euclidean cost | $\mathbf{C}_{ij}^{\star}$ MLE (1) | 100% | 2.09 | 3.23 | 3.99 | 4.43 | 4.95 |
| | | 83.3% | 2.06 | 3.19 | 4.11 | 4.47 | 4.83 |
| | | 66.7% | 2.19 | 3.47 | 4.14 | 4.55 | 4.88 |
| | $\mathbf{C}_{ij}^{\star}$ MLE (2) | 100% | 2.08 | 3.10 | 3.67 | 4.01 | 4.82 |
| | | 83.3% | 2.03 | 3.16 | 3.85 | 4.29 | 4.63 |
| | | 66.7% | 2.17 | 3.36 | 4.10 | 4.53 | 4.87 |
| | $\mathbf{C}_{ij}^{\star}$ MC (1) | 100% | 2.15 | 3.11 | 3.70 | 4.29 | 4.65 |
| | | 83.3% | 2.29 | 3.27 | 3.99 | 4.36 | 4.71 |
| | | 66.7% | 2.43 | 3.67 | 4.38 | 4.73 | 5.01 |
| | $\mathbf{C}_{ij}^{\star}$ MC (2) | 100% | 2.09 | 3.12 | 3.76 | 4.30 | 4.62 |
| | | 83.3% | 2.26 | 3.29 | 4.02 | 4.38 | 4.75 |
| | | 66.7% | 2.42 | 3.63 | 4.32 | 4.70 | 5.07 |
| Riem. cost | $\mathbf{C}_{ij}^{\star}$ MC (2) | 100% | 2.14 | 3.10 | 3.76 | 4.20 | 4.56 |
| | | 83.3% | 2.27 | 3.24 | 3.97 | 4.38 | 4.73 |
| | | 66.7% | 2.33 | 3.66 | 4.38 | 4.75 | 5.01 |

Table 1: **Comparison of constrained and unconstrained optimizations for synchronization of functional maps, for the MLE baselines and our proposed MC methods.** Experiments vary the variance $\sigma^2$ of the Gaussian noise defining the observed corrupted $\mathbf{C}_{ij}^{\text{corr}}$ within the objective function, and the sparsity of the synchronization graph on $N_s = 11$ cat shapes represented by the edge density $D$. The baselines MLEs are computed for the objective function defined with the Euclidean distance: MLE (1) is computed without constraining the optimization on $SO(n)$, while MLE (2) incorporates this constraint. The first proposed estimator MC (1) corresponds to the mean of the functional maps samples obtained from the Markov Chain in the MCMC, where the chain is not constrained to the Lie group $SO(n)$. The second proposed estimator MC (2) corresponds the mean of the samples from the Markov Chain in the MCMC, where the chain is constrained on $SO(n)$, computed either with Euclidean or Riemannian cost. The values reported correspond to the average Euclidean squared distance comparing each $\mathbf{C}_{ij}^{\star}$ to the corresponding ground truth $\mathbf{C}_{ij}^{\text{GT}}$ (lower is better). A visual representation of this data is shown in Figure 4.

Table 1 shows the comparison between the results obtained from MLE and the RLFM sampler for synchronization graphs on $N_s = 11$ cat shapes (nodes) and a varying edge density $D$. Experiments also vary the

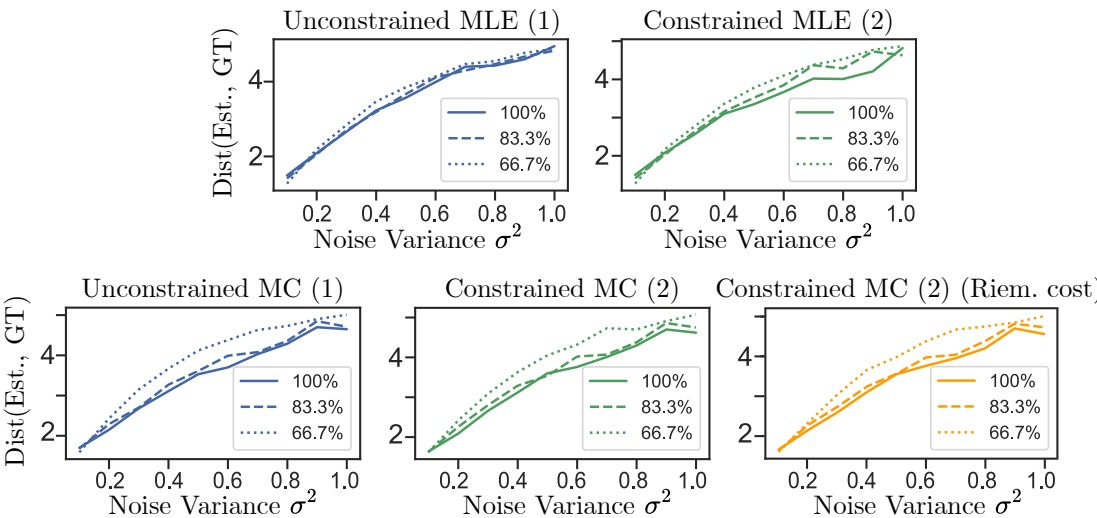

Figure 4: **Effect of graph density and data corruption level (noise variance) on the performances of the baselines MLEs and the proposed RLFMs.** Solid lines denote $100\%$ graph density, broken lines $83.3\%$ and dotted lines $66.7\%$. The X-axis represents the noise variance level, and the Y-axis is used to represent the Euclidean distance between estimated functional maps $\mathbf{C}_{ij}^{\star}$ and ground truth (GT) functional maps $\mathbf{C}_{ij}$, with the baselines and proposed methods as detailed in the main text.

variance $\sigma^2$ of the Gaussian noise defining the observed corrupted $\mathbf{C}_{ij}^{\text{corr}}$ within the objective function. A visual representation of this quantitative data is also shown in Figure 4.

Each value in this table refers to the Euclidean distance between the estimated output and ground truth functional map (lower distances are better). It can first be noted that the quality of the estimates worsens with the increase in corruption variance, which is expected as the optimization problem becomes increasingly challenging. We can see that outputs of the constrained MLE ($\mathbf{C}_{ij}^{\text{MLE (2)}}$) are closest to the ground truth, which empirically supports the geometric assumption we initially made and results from the previous subsection. However, we do not observe this trend with the outputs of the constrained MCs, which are comparable to their unconstrained variation.

Next, we reveal that the performance of the MLE estimators and the proposed RLFM estimators are comparable. This hints at the fact that the most important advantage of our proposed approach might not be accuracy, but rather the provision of a measure of uncertainty associated with predictions. The fact that the RLFM estimates do not outperform the MLE estimates can be accounted to the stochasticity associated with the RLFM sampler, which is known to approximate the exact posterior as the numbers of samples tend to infinity. This is supported by the observations obtained from the oracle method (Figure 3) where the overall accuracy increases with the increase in the number of samples. Interestingly, we observe on Figure 4 that our proposed RLFM estimates tend to leverage high connectivity of the synchronization graph, since higher graph edge density results in a drop of prediction error—a feature absent from the MLE prediction errors.

## 7 Limitations

The first assumption made in this paper is that the shapes are near isometric. This provides us with the convenient property of orthogonality among functional maps, and supports the geometric constraints introduced. However, this assumption might break for a range of surface meshes extracted from the real world. If the assumption breaks, then the Riemannian variations of our methods loose their mathematical grounding. This represents a limitation of our approach. Despite this, isometry is still a reasonable and

standard assumption used in computer vision and computer graphics literature due to its benefits. The cat bodies used in this paper are examples of non-rigid objects deforming in a nearly-isometric way.

The second limitation includes using special orthogonal groups such as $SO(n)$ to describe the underlying constraints of functional maps, instead of using the full Stiefel manifolds $St(n, n)$. This choice was made to simplify computational challenges. As a consequence, our approach cannot manage functional maps with negative determinants and it also prevents us from using rectangular functional maps, *i.e.* maps that match eigenbasis with different cardinalities on different shapes. Evaluating these would be a natural extension to the methods presented in this paper.

Finally, the Riemannian Langevin algorithm suffers from the well-known *meta-stability* phenomenon, i.e., the drawn samples concentrate near the global optimum if we wait for an exponential amount of time. In practice, our samples only characterize local minima.

## 8    Conclusion

In this paper, we developed a geometrically constrained probabilistic inference approach for the synchronization of functional maps used in shape correspondences. To this aim, we proposed and deployed a novel Riemannian MCMC sampler. We demonstrated the simultaneous performance of MLE estimation and Bayesian inference by sampling from the posterior using the MCMC. The key novelty of this work lies in uncertainty quantification (UQ) for functional map synchronization. The second contribution is to incorporate geometric constraints by optimizing functional maps in the Lie group $SO(n)$. We compared our proposed method with methods that do not restrict the functional maps on $SO(n)$, and with MLE estimators that do not provide uncertainty quantification across noise levels and connectivity of the synchronization graph. While baseline MLE estimators benefit from added geometric constraints, we observe that the MC estimators do not show improved performances in the data regimes that we investigated.

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

## A   Computation of Maximum-a-Posterior

We derive the formula for the Maximum a Posteriori (MAP) estimate of the absolute functional maps shown in Eq. (12). This MAP estimate follows from Bayes Theorem expansion for the maximum a-posteriori (MAP) estimate:

$$\mathbf{C}^\star = \arg\max_{\mathbf{C} \in \mathcal{F}^{N_s}} \frac{p(\mathcal{C} \mid \mathbf{C}) \times p(\mathbf{C})}{p(\mathcal{C})}. \tag{24}$$

The logarithm is a monotonic function which means that the argument maximum for the original and log-transformed optimization objective are identical. Taking the logarithm of both sides yields:

$$\mathbf{C}^\star = \arg\max_{\mathbf{C} \in \mathcal{F}^{N_s}} \Big( \sum_{(i,j)\in\mathcal{E}} \log p(\mathbf{C}_{ij}|\mathbf{C}) + \sum_{i=1}^{N_s} \log p(\mathbf{C}_i) - \sum_{(i,j)\in\mathcal{E}} \log p(\mathbf{C}_{ij}) \Big). \tag{25}$$

In the first term, we note that $C_{ij}$ only depends on $i$ and $j$. The third term in the above equation can be ignored due since we are optimizing only over $\mathbf{C} \equiv \{\mathbf{C}_i\}_{i=1}^{N_s}$. Assuming a Gaussian distribution over the group SO and assuming a uniform prior reformulates MAP into *Maximum Likelihood Estimation (MLE)* problem, as shown below:

$$\mathbf{C}^\star = \arg\max_{\mathbf{C} \in \mathcal{F}^{N_s}} \sum_{(i,j)\in\mathcal{E}} \log p(\mathbf{C}_{ij}|\mathbf{C}_i, \mathbf{C}_j). \tag{26}$$

