# OpenReview forum: "Probabilistic Riemannian Functional Map Synchronization  for 3D Shape Correspondence"
_TMLR — Withdrawn by Authors_

### Review · Reviewer_KaBE · 2023-02-09

**Summary Of Contributions:**

The paper is concerned with shape matching, i.e. given a series of meshes, find a matching between corresponding nodes in the meshes. This is phrased as a matching problem to a template (i.e. all meshes should be matched to a uniquely chosen mesh) as an optimization problem over the space of rotation matrices. This is a restriction over a previously developed idea of optimizing over the Stiefel manifold. It was unclear to me if this formulation (optimizing over rotations) is novel or if this has previously been done.

As the space of rotations is compact, the paper proposes uniform priors and a Riemannian Gaussian likelihood, such that maximum a posteriori estimation can be done with standard Riemannian optimization tools. To explore the uncertainty of the recovered mode, a Riemannian MCMC sampling algorithm is proposed.

Experiments on a small subset of the Tosca dataset suggest that the method is realizable in practice.

**Audience:**

Yes

**Claims And Evidence:**

Yes

**Requested Changes:**

Let me first say that I think the proposed methodology has merit and is of potential interest to a subset of the machine learning community. I do still have reservations, which the below changes could help settle.

### Top-most important changes
1. I struggle to determine methodological novelty, and I request a more clear picture of what is actually novel in the paper. In particular, with respect to the past work of Tolga Birdal and colleagues as this seems to be the foundation of the present work.
2. I miss either a wider selection of baselines (see Weaknesses above) or a more clear explanation of such is not relevant or another way for me to understand the strengths and weaknesses of the present model over existing work.

## Minor changes (but also easy ones)
I'd like to see the following questions answered (either in this forum or in the paper depending on where it is more appropriate):

3. In the second paragraph of Sec. 1 there is some confusion (to me) about if X consists of N or M points, i.e. if it has the same cardinality as Y.
4. In the same paragraph, should $X = P Y$ or rather $X \approx P Y$? To me, it seems the latter is more appropriate.
5. In the last paragraph of page 1, point (ii), it is argued that a difference in cardinality is a problem in the relaxed problem of finding $S$. Is this not also a problem in the original formulation of finding $P$?
6. On page 4, should the sentence "Consequently, it is possible to minimize (4).." instead refer to Eq. 5?
7. On page 6, what motivates the choice of an identity concentration matrix? Is it not important for subsequent Bayesian analysis that the likelihood is suitably scaled?

I would also like to see the following things improved upon:

8. To readers not familiar with shape matching, it might be beneficial to have a small illustrative example of the type of data, and the associated task. The textual description is currently somewhat abstract and can be hard to follow.
9. The first paragraph of page 2 is very brief, and I found it near-impossible to follow. It would be helpful (to this reader, at least) if this could be expanded somewhat.
10. In general, citations are formatted oddly. It would be good to replace \cite with \citet and \citep where appropriate.
11. I generally found the related work section to not be particularly informative. The main point currently seems to be to cite relevant work. It would be great if the section could be slightly tweaked to better emphasize the differences to the present work.
12. I did not understand the phrase "a canonical universe defining a canonical shape"; what is a universe in the current context?
13. On page 5: "Expecting that the true relative map is close to the expected value", what does that mean? Can this statement be formalized? In my understanding, this is an assumption that leads to Eq. 11. If so, a more precise statement of the assumption would be valuable.
14. In Sec. 6.2 it is observed that the MCMC sampler provides highly concentrated samples. This is presented as evidence of "stability". Could it also not simply mean that the sampler does not work? Or that the likelihood is poorly scaled (see question about identity concentration). Here I miss justification for the claim.
15. On page 13: it is not clear to me when negative determinants might appear; adding half a sentence describing such would be of value.
16. Is appendix A really needed? Unless I miss something, the statement proved is highly trivial.

**Strengths And Weaknesses:**

### Strengths:
* The paper is generally easy to follow (speaking as a reader familiar with Riemannian geometry).
* The paper does a nice job of discussing the limitations of the developed ideas.
* The idea of restricting to rotation matrices is reasonably well-motivated as it is significantly more practical than working with the Stiefel manifold.

### Weaknesses:
* The work appears highly incremental compared to the (many) cited papers from Tolga Birdal. I generally struggle to determine what is new in the paper, and to which extent the paper is instead a composition of previous work. I could use more clarity in the related work discussed throughout the article.
* Related to the previous comment, then it is unclear to which extent the proposed work improves over the previous work of Tolga Birdal. For example, the developed sampler appears near-identical to a previously developed sampler, yet no empirical comparison is performed.
* Related to the previous comment, then I miss a wider selection of baselines. Why is the Stiefel approach e.g. not compared to? Why no comparison to other sampling algorithms? Is it not possible to do MCMC in the unconstrained case? If so, why is such a comparison not performed? I find it difficult to determine how well the proposed methodology works (here I emphasize that it is not important to me if the developed method is the best, but I'd still like to know its empirical strengths and weaknesses).

---

### Review · Reviewer_sshG · 2023-02-15

**Summary Of Contributions:**

This paper introduces a new method to compute functional maps between multiple shapes while considering a cycle-consistency constraint and orthogonality constraints. Effectively, the authors propose a probablistic model and a sampler to produce these functional maps.


**Audience:**

Yes

**Broader Impact Concerns:**

this work seems niche in general, and I doubt it will have a broader impact outside the specific community of functional maps practitioners and researchers.

**Claims And Evidence:**

No

**Requested Changes:**

I would like to see an improved text of Sec. 4, properly describing the proposed model, as well as motivating its introduction with respect to existing work. The authors should also separate better existing work from novel work; this way, the reader will better understand the positioning of the paper in the context of previous work


**Strengths And Weaknesses:**

strength:
The authors claim to be the first to propose a functional map framework with uncertainty quantification abilities.

weaknesses:
unfortunately, the proposed model is unclear, both from a theoretical viewpoint and from a practical prespective. For instance, Eq. 14 and 15-18 seem to be representing the same ideas, but I might be wrong. In addition, the distinction between the work of Birdal & Simsekli '19 is unclear. In its current form, Sec. 4 is written in a mixed fashion where previous results are intertwined with potentially new results. Are the novel components in your work are Eqs. 21-22? How are these derived? are these known results from the literature on geodesic integrator or your own novel formulation? what are the theoretical differences with respect to Birdal & Simsekli '19? what about the practical differences?

---

### Review · Reviewer_rQgi · 2023-03-07

**Summary Of Contributions:**

This paper is about a probabilistic approach to 3D shape correspondence.
Shapes are represented via their coefficients with respect to a function basis, here, the Laplace-Beltrami eigenfunctions. The correspondence of two shapes can be modeled as an element of the Stiefel manifold ST(n,n) where n is the basis dimension, assumed the same in the two shapes. Instead of using the Stiefel manifold, an SO(n) rotation is applied. When multiple correspondences have to be synchronized, the cycle consistency constraint has to be optimized with respect to a multidimensional rotation. Optimization can be accomplished via the theory of optimization on Riemannian manifolds. However, the paper's main point is to formulate the synchronization problem probabilistically. This is done in two ways: one is to solve for the maximum a posteriori estimate that yields a maximum likelihood given a uniform prior. The other is to sample by applying the Riemannian Langevin Monte Carlo method. This results in a first order SDE that is integrated using the retraction Euler integrator.
Limited experiments on a set with four shapes are shown.

**Audience:**

Yes

**Broader Impact Concerns:**

No broader impact section is needed.

**Claims And Evidence:**

No

**Requested Changes:**

- The paper has to be rewritten by expanding and clarifying the probabilistic approach even if that means to immitate more the structure other papers like Birdal and Simsekli 2019
or the one by Chiuso et al. (see above).

- Literature and intro should focus on probabilities on groups rather than describing functional maps or cycle consistency that has been done in many other papers.

- Authors have to define exactly what they mean by uncertainty.

- Authors have to provide a quantitative evaluation of the uncertainty. One of the best cases is to address a dataset of objects with an inherent plane or point symmetry.

- Authors have to explain how well the uncertainty reflects the actual reality of wrong correspondences.

- A dataset with tens of objects should be used and the improvement on correspondence through cycle consistency has to be shown when the whole flexed MC approach is used.


**Strengths And Weaknesses:**

+ A probabilistic approach for correspondence synchronization is an open challenge.
A viable solution would enable multiple hypotheses on ambiguous correspondence cases.

+ An uncertainty quantification would give level of confidence and possibly guide us into which cases cycle consistency could resolve.

+ Experiments show the superiority of the Riemannian vs the Euclidean approach.

+ Experiments show the superiority of the MC sampling vs an MLE approach.

- It is not clear what is the novel contribution of the paper. The equations are taken verbatim from Birdal and Simsekli CVPR paper with a minor adaptation for SO(n) instead of the Birkhoff polytope. That CVPR paper was better structured with a rigorous organization.
A lot of insights about the SO(n) can also be gained from reading Wang et al. 2020.

- There is extensive literature in distributions on SO(n), with the best probably being  WIDE-SENSE ESTIMATION ON THE SPECIAL ORTHOGONAL GROUP, ALESSANDRO CHIUSO, GIORGIO PICCI, AND STEFANO SOATTO that has a comprehensive analysis including the SDEs of this paper.

- Problem 1 uses the tools from Pennec 2006, and while useful for a probabilistic interpretation of the solution $C_i^{+} C_j$ it does not help in quantifying uncertainty.

- At no point in section 4 do we see a definition of uncertainty. If it is the probability itself, there is a big question about the normalization with the total probability, as in many generative approaches. Is it the computation of the variance?

- The motivation of the paper is to handle uncertainty, which is a concept referring per definition to reality. However, the experiments use a ground-truth functional map, while the actual situation would be to introduce noise in the vertices of the mesh. The noise added to the functional maps does not have any grounding.

- The main point of probabilistic approaches is not to handle noise but to handle ambiguity. Noise can be handled with "least squares" (Problem 1), but ambiguity in the correspondence can be handled only with uncertainty. This is shown only in one anecdotal example in Figure 2.

- It is not clear how well-calibrated is the estimate of the uncertainty.

- Using only four instances of a shape defeats the purpose of cycle consistency. Not clear why so few.

---

### Review · Reviewer_Stgv · 2023-03-09

**Summary Of Contributions:**

The paper considers the problem 3D shape matching: it considers the framework of functional maps and models the correspondance of the given two shapes as the special orthogonal group. Similar to (Birdal & Simsekli, 2019), the paper proposes a probabilistic model for synchronization for 3D shape correspondence. The paper propose a sampler to perform the associated uncertainty quantification.

**Audience:**

Yes

**Claims And Evidence:**

No

**Requested Changes:**

The quantitative experiments requires a complete overhaul.

**Strengths And Weaknesses:**

1. Aims to learn the functional map by taking into account the geometry of the underlying space

2. While several qualitative results are shown of the proposed method, it is unclear if/how the proposed method improves upon the existing methods. For instance, Table 1 shows MLE methods are better than the proposed MC methods. Overall, the experiments seem to be quite weak and there does not seem to have any evidence to support the claims of the paper.

3. Is the information given in Table 1 simply replicated in Figure 4 or does Figure 4 have any additional information?

Minor comments:
1. \citep and \citet has been used interchangeably.

---

### Review · Reviewer_3TTH · 2023-03-12

**Summary Of Contributions:**

In this paper, the authors study the synchronization of functional maps for shape correspondences. The problem may be formulated as a constrained optimization problem on the Stiefel manifold. This paper proposes a probabilistic model, where the problem is formulated as a MAP estimation on the Lie group. Such probabilistic approaches were also explored in the literature. A Riemannian Langevin functional map sampler is further introduced to perform uncertainty quantification.

**Audience:**

Yes

**Claims And Evidence:**

No

**Requested Changes:**

1. The authors should make clear how well the proposed method works and when the readers can apply the proposed method in place of the baselines for better performance.

2. The authors should make clear the benefits and drawbacks of introducing geometric constraints in practical applications.

3. The authors should make explicit the primary contributions of this paper and go into more depth on the novelty in comparison to previous work.


**Strengths And Weaknesses:**

Strengths and Weaknesses:
1. This paper discusses the limitations of the proposed approach in detail.

2. The authors develop a probabilistic approach for synchronization on functional maps, which incorporates the geometric constraints defining the properties of functional maps, indicating that the functional maps are restricted on the Lie group.

3. It is unclear how the proposed method with geometric constraints outperforms the MLE baselines in numerical simulations.

4. It is unclear what the key contribution of this paper is and how it differs from previous work in terms of innovation.

---

### Note · Authors · 2023-03-20

**Comment:**

We thank the AE and the reviewers for their time, as well as their very constructive and valuable comments. We are withdrawing the submission to make time to put together a major revision of our work, taking the comments into account.

**Withdrawal Confirmation:**

I have read and agree with the venue's withdrawal policy on behalf of myself and my co-authors.